# Botulinum Toxin in Pain-Related Post-Stroke Limb Spasticity: A Meta-Analysis of Early and Late Injections

**DOI:** 10.3390/toxins17050258

**Published:** 2025-05-21

**Authors:** Frances Marie Tamayo, Raymond Rosales, Jörg Wissel, Bo Biering-Sørensen, Joshua Nathaniel Ellano, David Simpson

**Affiliations:** 1Department of Neuroscience and Behavioral Medicine, University of Santo Tomas Hospital, Manila 1015, Philippines; 2Research Center for Health Sciences, Faculty of Medicine and Surgery, University of Santo Tomas, Manila 1015, Philippines; 3Department of Neurorehabilitation and Physical Therapy, Vivantes Hospital Spandau, 13585 Berlin, Germany; wissel@rehaklinik-beelitz.de; 4NeuroScience Centre, Rigshospitalet, Copenhagen University Hospital, Havnevej 25, DK-3100 Hornbæk, Denmark; soerensen@regionh.dk; 5Department of Nursing, College of Nursing, University of Santo Tomas, Manila 1015, Philippines; 6Department of Neurology, Icahn School of Medicine at Mount Sinai, New York, NY 10029, USA; david.simpson@mssm.edu

**Keywords:** spasticity, pain-related post-stroke spasticity, botulinum toxin, early injection, late injection

## Abstract

Spasticity is a common complication associated with stroke, and around 72% of stroke patients will develop pain during the disease. Botulinum toxin (BoNT) is a safe and efficacious treatment for spasticity and can improve associated complications, including pain. Hence, this meta-analysis aims to establish whether BoNT can reduce pain-related post-stroke spasticity (pPSS) in the early treatment period (<12 weeks post-stroke) or in the late period (>12 weeks post-stroke) based on the available evidence. This study also aims to establish the dose–response relationship of BoNT-A in pPSS. Based on pooled data from multiple studies, there is no significant difference in the scores measuring pPSS between patients who received early BoNT-A injections and those who received a placebo. This finding suggests that within the early treatment period, BoNT-A may not be more effective than a placebo in reducing pPSS. However, it is important to note that the data for early BoNT-A injections are limited, indicating that research is needed to draw definitive conclusions [*z* = 3.90 (*p* < 0.0001)]. While BoNT-A appears somewhat more effective than a placebo in the late phase, as indicated by the small to moderate positive z value, there is not enough evidence to confidently claim superiority over a placebo [z = 1.48 (*p* = 0.14)].

## 1. Background

Stroke is one of the main causes of morbidity that affects an individual’s quality of life. Globally, one in four people over the age of 25 will have a stroke in their lifetime. Over 143 million years of healthy life is lost each year due to stroke-related death and disability. [1]. Spasticity, which occurs in many post-stroke patients, can be defined as disabling if it is perceived by the individual or caregivers as hindering a person’s current body function, activities, and participation [2]. In a prevalence study by Schinwelski et al. [3], patients with post-stroke spasticity (PSS) had worse Health-Related Quality of Life (HRQoL) in terms of physical functioning, role limitations, physical pain, and vitality. The pooled prevalence of spasticity after stroke was 25.3% [4]. Prevalence data on the different phases of the PSS continuum revealed spasticity rates of 4% to 27% in the early time course (week 1 to week 4), 19% to 26.7% in the post-acute phase (first to third month), and 17% to 42.6% in the chronic phase (third month) [5]. The onset of spasticity is variable in the post-stroke period, which usually develops and peaks at 1 to 3 months after a stroke. Again, in the same metanalysis by Zeng, clinically significant PSS was defined as having a Modified Ashworth Scale (MAS) score of ≥1, and disabling or severe spasticity was defined as having an MAS score of ≥3 [4]. Although the neuronal components of spasticity peak at 3 months after stroke, the muscular components of spasticity may increase over time, thus contributing to increased incidence of spasticity at 6 months post-stroke [6]. These timelines have been applied by many investigators in an attempt to effectively treat and prevent the progression of PSS, which affects limb mobility and flexibility. This is in accordance with the publication of European Expert Consensus on improving patient selection for the management of disabling spasticity, in which a consensus among 79 spasticity experts from 17 different countries recommends that BoNT-A should be initiated immediately or not more than 3 months after the onset of disabling spasticity to prevent the negative consequences of spasticity [2].

Chronic spasticity reduces the number of sarcomeres and can cause damaged muscles to have an increased proportion of connective tissue; these changes in the muscle matrix and soft tissue cause the pulling forces to be transmitted more readily to the muscle spindles [7]. The stretching of a spastic and contracted muscle can disrupt muscle fibers and release substances that excite the muscle nociceptors, which can ultimately be interpreted as nociceptive pain [8].

Persistent hypertonia, which is a result of the sustained contraction of muscles near and around blood vessels, can result in decreased oxygen availability or hypoxia. Hypoxia is a trigger for the release of inflammatory substances that can activate muscles’ nociceptive receptors, thereby promoting the pain cycle. At the same time, persistent hypertonia likely causes the level of oxygen consumption to increase, thereby worsening the pain situation. Muscle overactivity can also cause secondary nociceptive pain through abnormal loading on soft tissue and joints that also undergo structural changes during PSS [9].

In a recent scoping review conducted by Suputtitada and colleagues in 2024, the authors identified eleven modalities as treatment for spasticity of stroke, including repetitive peripheral magnetic stimulation (rPMS), non-invasive brain stimulation (NIBS), whole-body vibration (WBV), and BoNT-A injections [10]. Botulinum toxin (BoNT) is an established and effective treatment for spasticity in the subacute and chronic stages of stroke that addresses limb muscle hypertonicity, disability, and passive limb functions. BoNT-A is predominantly used to treat focal spasticity, and in clinics, this intervention complements neuro-rehabilitation practices [11]. BoNT-A can also have an analgesic effect; however, the mechanisms behind the analgesic effects of BoNT-A in humans are still unclear [12].

Owing to both the motor and sensory modulation effects of BoNT-A, its therapeutic applications in neuro-rehabilitation are geared toward not only reducing muscular hyperactivity but also providing relief from deranged sensory motor networks that result from post-stroke lesions. Studies on the early administration of BoNT-A in PSS are generally rationalized along the lines of contracture prevention and a possible delay in the timing of the re-injection of BoNT-A [13]. Regarding the optimal timing for targeting flexor muscles, within a period of 4–6 weeks after a stroke has been shown to be the most effective [10]. Compared to studies on reductions in muscle hyperactivity and disabilities, there are few available studies on BoNT-A’s effects in pPSS. In fact, the reduction of pPSS is also an important goal for targeted BoNT-A injections since this condition has a huge impact on quality of life and functioning. Hence, this present meta-analysis aims to establish whether BoNT-A can reduce pPSS in the early treatment period (less than 12 weeks post-stroke) or in the late treatment period (more than 12 weeks post-stroke) based on the available evidence. This study also aims to establish the dose–response relationship of BoNT-A in pPSS since several factors influence treatment options, including the severity of spasticity, the involvement of muscle groups (location), the stage of recovery (acute versus chronic phase), and even the medical condition of the patient.

## 2. Methods

For the conception of our clinical question, the PICO (population, intervention, comparison, outcome) framework was used. This meta-analysis included randomized clinical trials (RCT) that analyze pain-related limb spasticity in stroke patients. The risk of bias was assessed using the Cochrane risk-of-bias tool for randomized controlled trials [14]. This meta-analysis was registered on PROSPERO, an international platform of registered systematic review and meta-analysis protocols (Prospero ID number: 1044077).

## 3. Information Sources

Six databases were used for this meta-analysis, including PubMed, Google Scholar, ResearchGate, Cochrane, Medline, and EBSCO. Other references cited in the searched articles were evaluated as potential sources to be included in this meta-analysis. Cross-referencing was conducted when it was deemed necessary.

## 4. Search Terms and Assessment

The following keywords were used for this study: stroke, botulinum toxin, spasticity, and pain. In order to identify relevant studies in the databases, we employed Boolean operators within our search strings. ‘AND’ was used to combine essential concepts (e.g., ‘stroke’ AND ‘spasticity’ AND ‘BoNT-A’), while ‘OR’ broadened the search to include synonymous terms (e.g., ‘botulinum toxin’ OR ‘Botox’), alongside relevant subject headings.

The search included studies published from January 2000 to July 2024. Quality assessment for the yield articles utilized the Physiotherapy Evidence Database (PEDro) scale. The PEDro scale was created to determine the quality of clinical trials and consists of a checklist that assesses the internal validity, external validity, and interpretability of different clinical trials. A PEDro score of 8 out of 10 is considered optimal; hence, studies that had a score greater than 8 were included. Two independent raters assessed the yield studies after they were blinded to authors, journals, and results. The initial screening was performed by assessing the title and abstract of the yield studies. The inclusion criteria for this meta-analysis require the studies to be multicenter or single-center randomized controlled trials—either double-blind or single-blind—focused on stroke patients with pPSS, where the type, dosage in units (U), and administration schedule of BoNT-A are clearly specified, the timing of the stroke (ictus) is indicated, the pain measurement tools (such as the Visual Analogue Scale [VAS], Numeric Pain Rating Scale [NPRS], Disability Assessment Scale [DAS]-Pain or McGill Pain Questionnaire Short Form [MPQ-SF]) are described, and the analysis follows the intention-to-treat principle. The exclusion criteria for this study are as follows: non-randomized studies, editorial studies and descriptive studies, including case reports, case series, retrospective and prospective studies, studies examining spasticity as a result of other medical conditions (i.e., demyelinating diseases and trauma related diseases), and lastly, studies that primarily assessed other treatment modalities and only used BoNT-A injections as an adjunct.

The studies were further classified as early administration if BoNT-A injections were given within 12 weeks after the ictus of stroke and late administration if BoNT-A injections were given after 12 weeks post-ictus of stroke.

## 5. Characteristics of Studies and Synthesis of Results

Articles were screened from PubMed, Google Scholar, ResearchGate, Cochrane, Medline, and EBSCO. As seen in Figure 1, all studies selected for the review were RCTs. Initially, the search yielded two hundred eighty-seven (287) results from multiple sources. Cross references (*n* = 21), non-English text (*n* = 8), non-stroke, and non-BoNT-A interventions (*n* = 129) were automatically removed. Other reasons for the removal of records include difficulty with access, descriptive studies, including case reports and case series, cohort studies, and systematic reviews. Thirty studies were screened, and seventeen studies were removed because the analysis of pain was not correlated to PSS. Among the studies, two had no scores on pain that were specified, and one study used rehabilitation as the primary modality and BoNT-A as an adjunct. Twelve RCTs were analyzed, of which eight used *AbobotulinumtoxinA*, two used *OnabotulinumtoxinA*, one used *Neu-BoNT-A*, and one used *IncobotulinumtoxinA*, as illustrated in Figure 1.

## 6. Results

Table 1 shows that 12 studies were included in this meta-analysis. Three RCTs analyzed pPSS treated with early BoNT-A injections, and nine RCTs assessed late BoNT-A injections.

## 7. Analysis of Early BoNT Injections for Treating pPSS

The first study included for early BoNT-A injections in pPSS was conducted by Rosales and Asian cohorts (2012) [15], in which BoNT-A use was favored over a placebo with a mean difference of −0.14 (−0.30–0.02). Participants who received BoNT-A (*AbobotulinumtoxinA*) reported improvement in pPSS that was significant at weeks 4 and 24 (*p* = 0.0043 on week 4 and *p* = 0.0340 on week 24).

In contrast with the trial conducted by Seo and coworkers (2015) [16], which used the same dosage for both *OnabotulinumtoxinA* as the comparator and *Neu-BoNT-A* as the experimental drug, there was no significant difference in pPSS between the two groups (*p* < 0.001 by Wilcoxon signed-rank test). This finding is expected since all the participants included in the study both received BoNT-A. In another Asian study conducted by Rosales (2018) [17], no statistically significant difference was seen in the Global Assessment of Changes-Pain (GAC-pain) scores at the last visit of their assessment (*p =* 0.6128); however, numerically higher proportions of patients treated with BoNT-A descriptively disclosed improvements in their reported pain.

Figure 2 shows a total of three studies that investigated the effect of BoNT-A injections on pPSS within the early phase compared to a placebo. Across all studies, 206 participants were included in the BoNT-A (experimental) group, and 195 participants were included in the placebo (control) group. The overall mean difference between the experimental and control groups was 0.23 (95% CI: [−0.11, 0.35]), indicating a very small improvement in pain outcomes in favor of the control (z = 3.90, *p* < 0.0001).

## 8. Analysis of Late BoNT-A Injections for Treating pPSS

For late BoNT-A injections, the first included study was conducted by Kong (2007) [18], which showed reduction in the VAS of both the placebo and BoNT-A groups at weeks 4, 8, and 12; however, there is no statistical difference established between the two groups (*p* > 0.05). Marciniak and colleagues (2012) [19] conducted a study in which both the late BoNT-A and placebo groups showed reductions in pain ratings only at week 4 but, overall, showed no significant improvements in pain scores, with upper body dressing and pain interfering with sleep at weeks 2, 4, and 12 (*p* > 0.05). Marco and colleagues (2007) [20] also observed decreased pain, as indicated by VAS measurements during the first week after BoNT-A administration. The VAS mean scores showed significant improvements during the first, third, and sixth months, with a statistically significant difference between the placebo and BoNT-A groups (*p* = 0.035). Shaw and coworkers (2011) [21], however, did not find any significant differences in changes in pain scores from baseline to the first or third months (*p* = 0.600, *p* = 0.269). Nonetheless, a significant reduction in pain scale was seen from baseline to 12 months only in the intervention group compared to the control group (*p* = 0.004). Tan and colleagues (2021) [22] found no significant differences between the BoNT-A and placebo groups in terms of their VAS, MAS scores, or shoulder abduction and external rotation ROM for hemiplegic patients at week 24 (*p* > 0.05). Yelnik and colleagues (2007) [23] observed pain reduction from baseline to the fourth week, with a four-point improvement in the BoNT-A group versus a one-point improvement in the placebo group (*p* = 0.025).

Masakado et al. (2020) [24] showed that the change in DAS scores was greater with BoNT-A than a placebo. This change was significantly greater with 400 U of BoNT-A at week 4 (0.52 [0.184]; *p* = 0.0067), week 6 (0.53 [0.198]; *p* = 0.0097), and week 8 (0.40 [0.196]; *p* = 0.0476). The change versus the placebo with 250 U BoNT-A was greater at week 8 (0.53 [0.226]; *p* = 0.0254).

Lastly, Wissel and coworkers (2016) [25] conducted a randomized double-blind placebo-controlled trial, which showed that the mean pain reduction from baseline to week 12 was significantly greater in the BoNT-A group (0.77, 95% CI 1.14 to 0.40) compared to the placebo group (0.13, 95% CI 0.51 to 0.24; *p* < 0.05).

Figure 3 shows the late studies that evaluated the effect of BoNT-A on pPSS compared to a placebo. The analysis involved 528 participants in the BoNT-A (experimental) group and 457 in the placebo (control) group. The overall mean difference between the experimental and control groups was 0.11 (95% CI: [−0.04, 0.26]), showing improvement in pPSS favoring BoNT-A injections with a small to moderate effect (z = 1.48). However, this effect was not enough to claim superiority (*p* = 0.14). There was substantial heterogeneity among the studies (Chi^2^ = 82.59, *p* < 0.00001, I^2^ = 87%), suggesting significant variability in the findings.

Several studies had a major impact on the overall effect size due to their large sample sizes and significant weight within the meta-analysis: Wissel et al. (2016) [25] contributed the largest proportion to the analysis, with a weight of 51.6%. Masakado et al. (2020) [24] contributed significantly, with weights of 11.1% for the 250 mg dose and 13.4% for the 400 mg dose. Both doses demonstrated positive effects in favor of BoNT-A, with mean differences of 0.09 (95% CI: [−0.36, 0.54]) and −0.04 (95% CI: [−0.45, 0.37]), respectively. These studies provided some evidence that higher doses of BoNT-A may offer relief in pain management for spasticity, counteracting some of the findings from larger, more neutral studies. Shaw et al. (2015) [21], which had a weight of 9.2%, reported a negative mean difference of −1.40 (95% CI: [−1.89, −0.91]), indicating that a placebo was more effective than BoNT-A in reducing pain.

The dominance of Wissel et al. (2016) [25] in the analysis, contributing 51.6% of the total weight, shaped the overall effect, as the study reported differences between BoNT-A and a placebo. Conversely, Masakado et al. (2020) [24], which showed a positive effect for BoNT-A, suggests that under certain conditions—such as higher BoNT-A doses—there may be a benefit related to pain management in PSS.

A funnel plot is used to assess potential publication bias by plotting the standard error (SE) of the mean difference against mean difference values. The symmetry of the funnel plot is a key indicator of the absence of publication bias.

For the early BoNT-A injections, the funnel plot, as seen in Figure 4, appears asymmetrical, with fewer studies in the lower part of the funnel, which could indicate missing studies with negative or null effects. However, for late BoNT-A injections, as seen in Figure 5, the points are distributed relatively symmetrically around the mean difference, suggesting that there is no significant publication bias. This implies that studies with both positive and negative results have been included in the meta-analysis, which is crucial for obtaining an unbiased overall effect estimate.

## 9. Dose–Response Relationship of BoNT Injections and Pain Scores

In the study conducted by Bhakheit et al. (2000) [26], which looked into using three standard doses of *AbobotulinumtoxinA* (500 U, 1000 U, and 1500 U), no dose–response relationship was clearly established regarding pPSS. Instead, it was found that better outcomes with pain were achieved at 1000 U of *AbobotulinumtoxinA*, although this finding is still statistically non-significant. Mancini [27] analyzed three different doses of *OnabotulinumtoxinA* for treating pPSS (Group I was injected with doses of 25 U to 50 U, Group II was given 50 U to 100 U, and Group III was given 75 U to 200 U), and the results showed no significant differences among the three groups on the 4th and 16th weeks.

## 10. Discussion

Spasticity-related pain is derived from severe muscle spasms, co-contraction, abnormal posturing, and the setting-in of biomechanical forces along the musculoskeletal region of immobility. The relaxation of hyperactive muscles brought about by BoNT-A reduces pain through the decompression of the nerve fibers by virtue of a reduction in muscle tone or volume. This phenomenon may also be brought about by the reduced afferent activity of spindles and decreased excitability of motor neurons. Furthermore, when using BoNT-A to treat pPSS, there are issues that need to be resolved: (1) the injected dosage: it is unknown whether a lower BoNT-A dose, in contrast with normal dosing, can achieve sufficient pain reduction; (2) duration of action: it is unknown whether the effect of BoNT-A on pain relief goes beyond the duration of action observed in spasticity treatment; and (3) inoculation techniques: it is unknown whether intraarticular or intramuscular administration would provide better pain relief [13]. The synaptosomal-associated protein of 25 kDa (SNAP 25) is a crucial protein involved in neurotransmitter release at the synapses. It plays a central role in the SNARE complex, a protein assembly that enables synaptic vesicles to fuse with the plasma membrane and release their neurotransmitter contents. By inducing the cleavage of SNAP 25, BoNT-A may also interfere with protein translocation from the endosomal compartment to the cell plasma membrane. It is also postulated that the ability of BoNT-A to decrease pain is due to the inhibition or interference with sodium (Na⁺) channel function related to neuropathic pain. The mechanism of BoNT-A at the neuromuscular junction is demonstrated in Figure 6 [28]. This mechanism is further supported by studies on the Transient Receptor Potential Vanilloid 1 (TRPV1) capsaicin receptor, an important non-selective cation channel involved in pain transmission [29]. Both of these mechanisms may be related to BoNT-A effects in non-muscle-based injections geared toward neuropathic pain.

By acting directly on peripheral nociceptive neurons, BoNT-A has been shown to reduce calcitonin gene-related peptide (CGRP), Glutamate, and Substance P. By reducing these pain-related neurotransmitters, central sensitization can be prevented. Nociceptive neurotransmitters act as mediators in the pathophysiology of pain, including neuropeptides that are also observed in disorders affecting peripheral structures such as tendons [30]. The number of CGRP muscle afferent neurons also increases 12 days after muscle inflammation [31]. Both of these inflammatory reactions are also responsible for pain generation. All these mechanisms are now postulated to contribute to the generation of pPSS, as explained in Figure 7.

The ability of BoNT-A to reduce muscle tone via the chemo-denervation of the injected hyperactive muscles and modulation of sensory motor programs is the rationale for the early use of BoNT-A. This rationale, as applied in early BoNT-A use, may not only prevent contractures but also prolong the needed re-injection intervals to control PSS [32]. Patients receiving routine BoNT-A treatments were also less likely to experience soft tissue shortening in the shoulder and had a lower incidence of the complete loss of proximal function compared to patients receiving routine BoNT-A treatments intermittently (3.9% vs. 20.8%) [33].

A total of 65% of patients with spasticity reported experiencing pain, with 60% characterized as aching pain. Most of the patients reported worse pain during movement (34%) than at rest (21%). It is also reported that the severity of pain and the severity of spasticity is not linearly correlated (r = 0.16; *p* > 0.05). A total of 80% of the patients believed that their pain was linked to spasticity, and around 62% noted a reduction in the severity of their pain following BoNT-A injections [34].

Woo and coworkers (2021) [35] analyzed the use of high-dose BoNT-A (*OnabotulinumtoxinA*, *IncobotulinumtoxinA*, and *AbobotulinumtoxinA*) in the early-start group (500 U and 600 U in the first three cycles), the dose of which was higher in contrast to that typically used in trials. The findings showed that higher doses of BoNT-A in the early-start group often resulted in longer intervals between needed re-injections than in the late-start group [36].

Similarly, in a review conducted by Intiso (2015) [13], high doses of up to 400 U of *OnabotulinumtoxinA* and *IncobotulinumtoxinA* and 500–1000 U of *AbabotulinumtoxinA* have been used to treat stroke-related upper limb spasticity, while doses of 300–400 U of *OnabotulinumtoxinA* and doses up to 1500 U of *AbobotulinumtoxinA* have been practically used to treat lower limb spasticity [13]. One RCT that primarily evaluated low doses was conducted by Mancini (2005) [27], in which there were no significant differences established.

A similar reluctance to inject the shoulder was seen at the international level; however, more recent results from the ULIS-III study showed an increase in injections, perhaps reflecting the recent increased education on the relevance of injecting these muscles [37]. It is important to note that the improvement in motor function relative to spasticity reduction is not the only primary end point for using BoNT-A; clinical parameters such as pain reduction and disability improvement should always be prioritized [36].

## 11. Limitations and Future Directions

To date, only a limited number of studies have examined the use of BoNT-A during the early phase of PSS, which leads to pain, among other effects. Even fewer studies have specifically analyzed its direct impact on pain, as most research focuses on its ability to alleviate pain indirectly by reducing spasticity. Future studies on PSS should focus on pain relief and try to differentiate between the circumstances eliciting the pain in addition to dividing the pain conditions into nociceptive pain, neuropathic pain, and nociplastic pain. Investigating the mechanisms underlying pain may lead to more effective and individualized pain treatments. BoNT-A might theoretically impact all pain conditions; hence, phenotyping a patient’s pain condition will be essential in future studies.

## 12. Conclusions

Although a favorable trend is emerging, pooled data suggest that within the early treatment period (<12 weeks), BoNT-A may not be more effective than a placebo in reducing pPSS. This study also concludes that during the late treatment period (>12 weeks), BoNT-A is still preferred over a placebo for the management of pPSS. There is still a lack of clear data on the dose–response relationship of BoNT injections for pain associated with post-stroke spasticity.

## Figures and Tables

**Figure 1 toxins-17-00258-f001:**
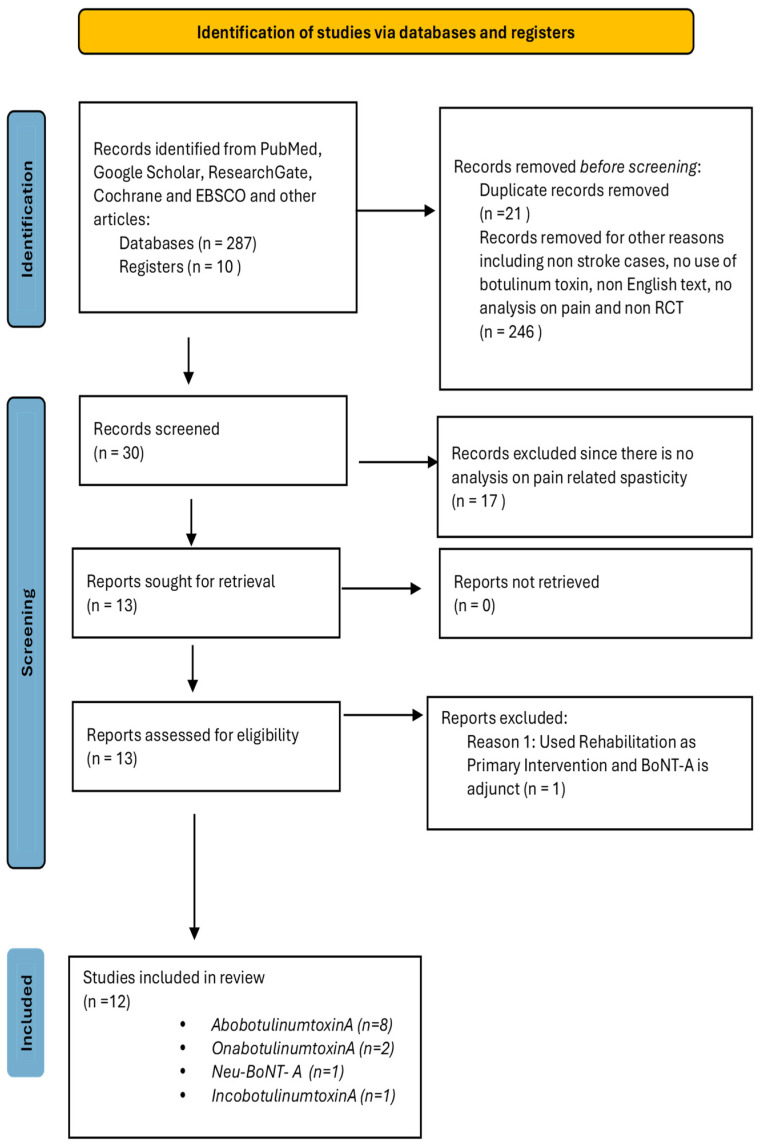
PRISMA flow diagram depicting search for BoNT-A studies on pain-related post-stroke spasticity (pPSS).

**Figure 2 toxins-17-00258-f002:**
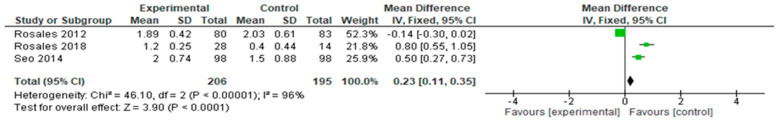
Early BoNT-A studies in pain-related post-stroke spasticity [15,16,17].

**Figure 3 toxins-17-00258-f003:**
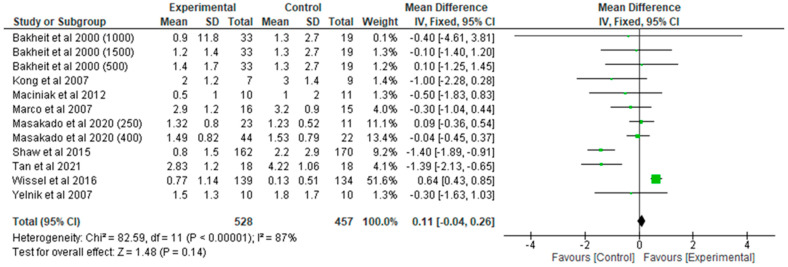
Late BoNT-A studies on pain-related post-stroke spasticity [18,19,20,21,23,24,25,26].

**Figure 4 toxins-17-00258-f004:**
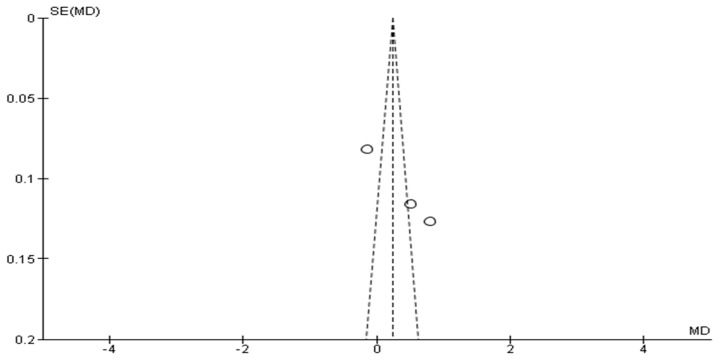
Funnel plot of early BoNT-A injection studies.

**Figure 5 toxins-17-00258-f005:**
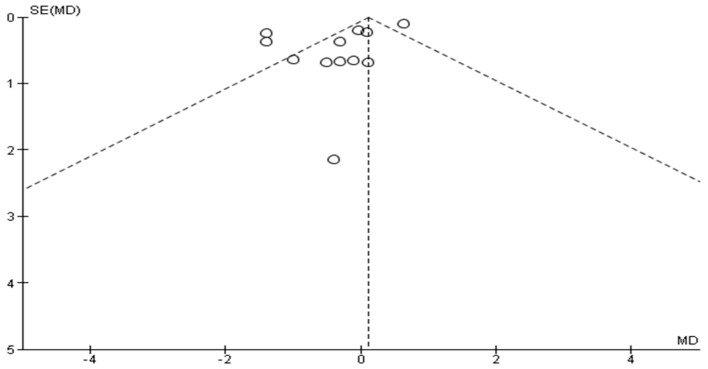
Funnel plot of late BoNT-A injection studies.

**Figure 6 toxins-17-00258-f006:**
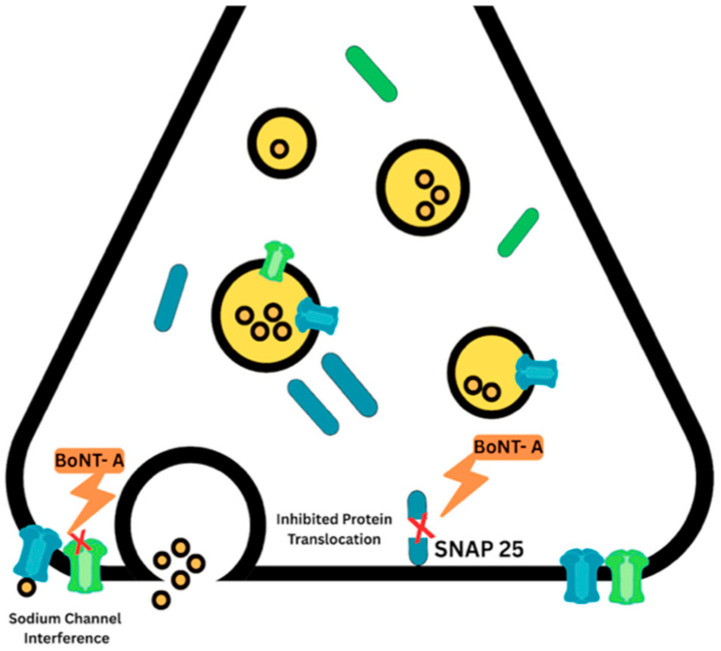
BoNT-A mechanism of action in the neuromuscular junction. BoNT-A causing sodium (Na+) channel interference and the cleavage of SNAP 25, resulting in disruption of translocation of important proteins that promote pain generation.

**Figure 7 toxins-17-00258-f007:**
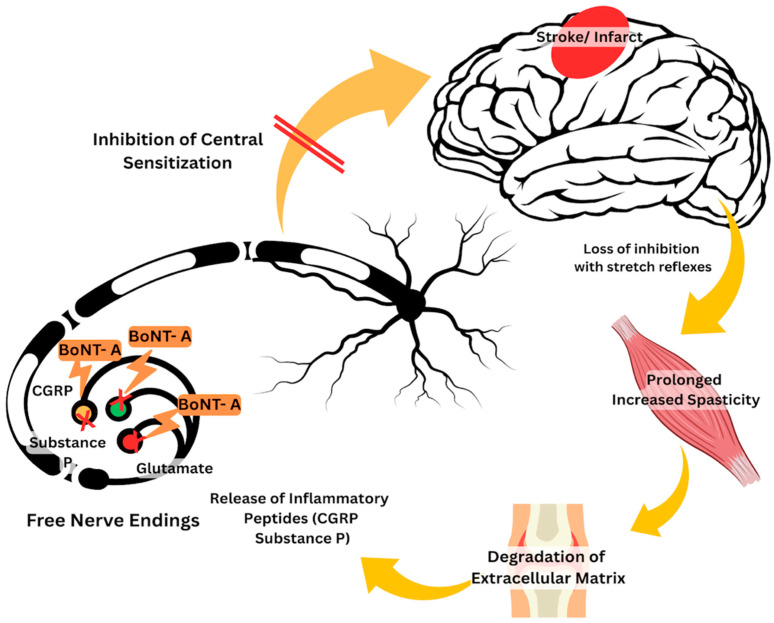
Illustrated mechanism of BoNT-A on proposed pathophysiology of pPSS. In patients with stroke, the loss of inhibition of the stretch reflexes in the upper neurons results in spasticity. It is postulated that spasticity in prolonged cases can induce inflammation. Studies have shown that repeated inflammation can cause the release of peptides, including calcitonin gene-related peptide (CGRP) and Substance P. BoNT-A acts on the peripheral blockade of the CGRP, Substance P, and Glutamate receptors, which ultimately inhibits the central sensitization of pain.

**Table 1 toxins-17-00258-t001:** Early BoNT-A and late BoNT-A injection studies included in the meta-analysis.

Author (Year)	Level of Evidence	Population	Ictus	Botulinum Injection Dose	Control	Pain Scale Applied
**Early Injections (Less than or Equal to 12 weeks)**						
Rosales et al. (2012)	Multicenter Randomized Placebo-Controlled Trial	163 patients with mild to severe upper limb spasticity	12 weeks	*AbobotulinumtoxinA* 500 U uniform dose	2.5 mL of Normal saline	VAS
Rosales et al. (2018)	Randomized Double-Blind Placebo-Controlled Trial	42 patients with upper limb spasticity	2–12 weeks	*AbobotulinumtoxinA* 500 U uniform dose	2.5 mL of Normal Saline	PNRS
Seo et al. (2015)	Randomized Controlled Trial	196 patients with moderate to severe upper limb spasticity	6 weeks	*Neu-BoNT-A* 10 U–360 U	BoNT-A 100 U	DAS-Pain
**Late Injections (More than 12 weeks)**						
Bakheit et al. (2000)	Randomized Double-Blind Placebo-Controlled Dose-Ranging Trial	83 patients with moderate to severe upper limb spasticity	>3 months	*AbobotulinumtoxinA* 500 U, 1000 U and 1500 U	2 mL of Normal saline	4-Point Pain Scale ^a^
Kong et al. (2007)	Randomized Double-Blind Placebo-Controlled Trial	17 patients with shoulder spasticity	>3 months	*AbobotulinumtoxinA* 250 U and 500 U	2.5 mL of Normal saline	VAS
Marciniak et al. (2012)	Randomized Double-Blind Placebo-Controlled Trial	21 patients with shoulder spasticity	28.8 months	*AbobotulinumtoxinA* 100 U and 150 U	1–1.5 mL of Normal Saline	MPQ-SF
Marco et al. (2007)	Double-Blind Randomized Clinical Trial	14 patients with shoulder spasticity	>3 months	*AbobotulinumtoxinA* 500 U	2.5 mL of Normal Saline	VAS
Shaw et al. (2015)	Multicenter Parallel Group Randomized Placebo-Controlled Trial	333 patients with upper limb spasticity	>1 month	*AbobotulinumtoxinA* 100 or 200 U	Standard Upper Limb Therapy	Pain Score Pain Description
Wissel et al. (2016)	Multicenter Randomized Double-Blind Placebo-Controlled Trial	273 patients with upper and lower limb spasticity	>12 weeks	*OnabotulinumtoxinA* 340–365 U	Normal Saline	PNRS
Tan et al. (2021)	Randomized Double-Blind Placebo-Controlled Trial	36 Patients with shoulder spasticity	>2 months	*OnabotulinumtoxinA* 50 U-100 U	2.0 mL of Normal Saline	VAS
Yelnik et al. (2007)	Randomized Double-Blind Placebo-Controlled Two Parallel Group Study	20 patients with shoulder spasticity	224 days	*AbobotulinumtoxinA* 500 U	BoNT-A solvent	VAS
Masakado et al. (2020)	Randomized Double-Blind Placebo-Controlled Study	100 patients with unilateral post-stroke upper limb spasticity	>16 weeks	*IncobotulinumtoxinA* 400 U	Injection Diluent 50 U/mL	DAS

Abbreviations: BoNT-A: botulinum toxin type A, DAS: Disability Assessment Scale-Pain, VAS: Visual Analogue Score, MPQ-SF: McGill Pain Questionnaire-Short Form 4-Point Pain Scale. ^a^ 0 = no pain, 1 = mild pain, 2 = moderate pain, 3 = severe pain.

## Data Availability

No new data were created or analyzed in this study.

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
