# Peer review of "Botulinum Toxin in Pain-Related Post-Stroke Limb Spasticity: A Meta-Analysis of Early and Late Injections"

_toxins, 2025, doi:10.3390/toxins17050258_

Round 1
Reviewer 1 Report
Comments and Suggestions for Authors
Introduction
1. The introduction presents BoNT-A as a go-to treatment right from the start, without first laying out that the evidence is still mixed or limited in some areas.
2. It doesn’t clearly explain why it’s important to compare early and late treatment—there’s no strong case made for why this timing issue matters in real-world practice.
3. The background gives a lot of information, but it doesn’t really question or critique previous studies, especially those with conflicting results or weak methods.
Methods
1. The review wasn’t registered in something like PROSPERO, which is a concern for transparency.
2. The review limited to English-language studies, so there’s a chance of other papers published in other languages were overlooked.
3. The search strategy are vague—there’s no list of the exact keywords or how they were used together, which makes it difficult for others to reproduce the search.
4. It’s not totally clear if all the included studies used intention-to-treat analysis, even though that was listed as a requirement.
5. There’s no breakdown of how the PEDro scores were applied, or whether study quality and bias were looked at closely.
6. Included studies used different BoNT-A types, dose, and muscle targets, but these differences aren’t really addressed or adjusted for.
7. 12-week cut-off used to define "early" vs. "late" treatment isn’t clearly justified—it feels arbitrary without a strong explanation.
Results
1. The results are pulled from studies that vary a lot (heterogeneity was really high), which makes the pooled conclusions less reliable, but this isn’t discussed much.
2. The review tries to look at different dosages, but the results are murky, and it doesn’t do much to make sense of them.
3. Only a few studies focused on early injections, which makes any firm conclusions about early treatment pretty weak.
4. Some large studies carry a lot of weight in the analysis, which skews the overall picture and makes it less balanced.
5. There’s no look at how different types of patients (age, health conditions, etc.) might respond differently to the treatment.
6. Both the placebo and BoNT groups showed some pain reduction, but the role of placebo isn’t really discussed, which is a big miss in a pain-focused review.
Discussion
1. The discussion spends a lot of time on biological details—receptors and mechanisms—but doesn’t connect that back to what clinicians can do with the info.
2. The focus is entirely on pain scores, with no mention of how treatment affects function, mobility, or everyday life.
3. Some non-significant results are covered which supported BoNT-A, which could be misleading.
4. There’s no attempt to break down the types of pain , even though that’s important for treatment decisions.
5. The weaknesses in the included studies (small sample sizes, different outcome measures, etc.) aren’t explored or critiqued.
Conclusion
1. The conclusion makes it sound like BoNT-A clearly works better in late-stage treatment, even though the stats don’t support that.
2. It downplays the limitations discussed earlier
3. It misses a chance to offer practical advice
Author Response
Dear Reviewer,
We would like to express our sincere appreciation for the insightful comments and constructive suggestions provided regarding our manuscript entitled “Botulinum Toxin in Pain Related Post Stroke Limb Spasticity: A Meta-Analysis on Early and Late Injections”.
Your feedback has been extremely valuable in improving the clarity and overall quality of our work. We have carefully considered all points raised and have revised the manuscript accordingly. Here, we provide our responses to each reviewer, addressing their comments in a point-by-point manner.
We are grateful for the time and effort invested by the reviewers and yourself in the evaluation of our manuscript, and we hope that the revised version meets your expectations.
Thank you for your continued consideration.
Sincerely,
On behalf of all co-authors
Introduction
Comment 1: [ The introduction presents BoNT-A as a go-to treatment right from the start, without first laying out that the evidence is still mixed or limited in some areas.]
Response 1: Thank you for this. This was addressed on LINE 62-65. “In a recent scoping review done by Suputtitada and colleagues in 2024, they identified eleven modalities as treatment for spasticity of stroke including Repetitive peripheral magnetic stimulation (rPMS), Non-invasive brain stimulation (NIBS), Whole-body vibration (WBV) and Botulinum toxin A (BoNT-A) injection”
Reference: Suputtitada A, Chatromyen S, Chen CPC, Simpson DM. Best Practice Guidelines for the Management of Patients with Post-Stroke Spasticity: A Modified Scoping Review. Toxins (Basel). 2024 Feb 10;16(2):98. doi: 10.3390/toxins16020098. PMID: 38393176; PMCID: PMC10892074.
Comment 2: [It doesn’t clearly explain why it’s important to compare early and late treatment—there’s no strong case made for why this timing issue matters in real-world practice.]
Response 2: This was addressed on LINE 76-77 “Optimal timing to target the flexor muscles within a period of 4–6 weeks after a stroke have been shown to be the most effective. “
Reference: Suputtitada A, Chatromyen S, Chen CPC, Simpson DM. Best Practice Guidelines for the Management of Patients with Post-Stroke Spasticity: A Modified Scoping Review. Toxins (Basel). 2024 Feb 10;16(2):98. doi: 10.3390/toxins16020098. PMID: 38393176; PMCID: PMC10892074.
Comment 3: The background gives a lot of information, but it doesn’t really question or critique previous studies, especially those with conflicting results or weak methods.
Response 3: The study critically reviewed many RCTs, resulting in the exclusion of several that investigated BoNT in combination with other treatment modalities (e.g., physical therapy), as these could act as confounding variables. As a group of researchers, we advocate for further investigation into the optimal timing of intervention to prevent the sequelae of post-stroke spasticity, particularly pain.
Methods
Comment 1:[The review wasn’t registered in something like PROSPERO, which is a concern for transparency.]
Response 1: This study was registered in PROSPERO. This addressed on LINE 92-93
Comment 2: [The review limited to English-language studies, so there’s a chance of other papers published in other languages were overlooked.]
Response 2: We acknowledge that the exclusion of studies published in languages other than English is a potential limitation of this study.
Comment 3: [The search strategy are vague—there’s no list of the exact keywords or how they were used together, which makes it difficult for others to reproduce the search.]
Response 3: This was addressed on LINE 102-106. The following keywords used for this study were Stroke, Botulinum Toxin, Spasticity and Pain. In order to identify relevant studies in databases, we employed Boolean operators within our search strings. 'AND' was used to combine essential concepts (e.g., 'stroke' AND 'spasticity' AND 'BoNT-A'), while 'OR' broadened the search to include synonymous terms (e.g., 'botulinum toxin' OR 'Botox'), alongside relevant subject headings.
Comment 4: [It’s not totally clear if all the included studies used intention-to-treat analysis, even though that was listed as a requirement.]
Response 4: All the included RCTs were reviewed for intention to treat and all the studies had this requirement.
Comment 5: [There’s no breakdown of how the PEDro scores were applied, or whether study quality and bias were looked at closely.]
Response 5: This was addressed on Line 113-115. A PEDro score of 8 out of 10 is considered optimal hence studies that had a score greater than 8 were included. Two independent raters assessed the yield researches to which they were blinded with authors, journal, and results.
Comment 6: [Included studies used different BoNT-A types, dose, and muscle targets, but these differences aren’t really addressed or adjusted for.]
Response 6: This will be a declared limitation for the study.
Comment 7: [12-week cut-off used to define "early" vs. "late" treatment isn’t clearly justified—it feels arbitrary without a strong explanation.]
Response 7: These timelines have been explained in the background: Although the neuronal components of spasticity peak at 3 months after stroke, the muscular components of spasticity may increase over time, thus, contributing to increased incidence of spasticity at 6 months post-stroke. [6] These timelines have been applied by many investigators in an attempt to effectively treat and prevent the progression of PSS which affects limb mobility and flexibility. This is in accordance with the publication of European Expert Consensus on improving patient selection for the management of disabling spasticity where there was consensus among the 79 spasticity experts from 17 different countries that BoNT-A should be initiated immediately or not more than 3 months after the onset of disabling spasticity, in order to prevent negative consequences of spasticity. [2]
Results
Comment 1: [ The results are pulled from studies that vary a lot (heterogeneity was really high), which makes the pooled conclusions less reliable, but this isn’t discussed much.]
Response 1: The high level of heterogeneity observed among the included studies is acknowledged as a limitation that may affect the reliability and generalizability of the pooled estimates.
Comment 2: [The review tries to look at different dosages, but the results are murky, and it doesn’t do much to make sense of them.]
Response 2: Regarding the analysis of different dosages of the injections, the available data were limited and inconsistently reported, which contributed to the murkiness of the findings. Hence, we will recommend that future trials adopt uniform dosing protocols and report dosage-specific outcomes to better elucidate the efficacy and safety profiles across dosage regimens.
Comment 3: [ Only a few studies focused on early injections, which makes any firm conclusions about early treatment pretty weak.]
Response 3: A recognized limitation of this study is the current scarcity of research focusing on early injections. We hope this review will encourage future researchers to explore this area further, as it is a key focus of our advocacy.
Comment 4: [ Some large studies carry a lot of weight in the analysis, which skews the overall picture and makes it less balanced.]
Response 4: Yes, the studies are heterogenous given that we are still investigating a topic with a limited data set. Hopefully this review will lead direction in analysis of BoNT in the context of stroke, spasticity and pain. Hopefully, this will also open doors to investigating BoNT for pain management directly and not as an indirect consequence of spasticity management.
Comment 5: [ There’s no look at how different types of patients (age, health conditions, etc.) might respond differently to the treatment. ]
Response 5: Agree to this comment, we can no longer restrict our studies to a certain age or sex but it is most definitely that the included patients are stroke patients solely with no other neuromuscular conditions since this is a declared exclusion for this review.
Comment 6 : [Both the placebo and BoNT groups showed some pain reduction, but the role of placebo isn’t really discussed, which is a big miss in a pain-focused review.]
Response 6: Thank you for this; The placebos are identified in the summary table.
Discussion
Comment 1: [The discussion spends a lot of time on biological details- receptors and mechanism but does not connect to what clinicians can do with the info.]
Response 1: Yes, we do agree with this that we went down on discussing the receptors and mechanisms since we are trying to explain/ postulate that BoNT- A does have a pain reducing effect independent on its own.
Comment 2: [ Focus is entirely on pain scores with no mention of how treatment affects function, mobility, or everyday life.]
Response 2: We have to agree with this comment; but including these other parameters as well will include a lot of analysis of factors including the psychological state of the patient/s with their perception of pain.
Comment 3: [Some non-significant results are covered which supported BoNT-A, which could be misleading.]
Response 3: Yes, we have to include these for the purpose of discussion.
Comment 4: [There's no attempt to break down the types of pain, even though that's important for treatment decisions.]
Response 4: Thank you for this. This is our attempt to look into pain first in general; the phenotyping of pain is a large discussion when it comes to neurologic conditions like stroke hence this is one of our recommendations.
Comment 5: [The weaknesses in the included studies (small sample sizes, different outcome measures, etc.) aren't explored or critiqued.]
Response 5: Thank you for this. We declared this as a limitation of our study.
Conclusion
Comment 1: [The conclusion makes it sound like BoNT-A clearly works better in late-stage treatment, even though the stats don't support that.]
Response 1: Yes, the p value was at 0.14; the forrest plot shows that it prefers experimental (BoNT-A) but yes agree with the p-value
Comment 2: [It downplays the limitations discussed earlier.]
Response 2: Agree with this.
Comment 3: [It misses a chance to offer practical advice]
Response 3: Agree with this, we are introducing a mechanism backed up by postulates, and the studies that are included are still very limited.

Reviewer 2 Report
Comments and Suggestions for Authors
This is an interesting study, but has not been adequately proof read. For example, sometimes 'BoNT-A' is used, and sometime 'BoNT- A', and sometimes spacing between words is inconsistent (e.g., line 49).
Abbreviations and acronyms used are not always defined, and should be when first used. A list of these would help some readers.
Line 95: Were ONLY RCTs included? If so, please indicate.
For example, BoNT-A, HRQoL
Methodology is good (use of PEDro; careful exclusions); however, with only 3 studies for early BoNT-A, how valid are the results?
Lines 170-1: 'not statistically significant (Z = 3.90, 170 P <0.0001).' But p is very small, so how is this non-significant?!xx
Methods and software used for Chi2 and I2, as well as other statistical tests, should be more fully described. Was RevMan used, for example?
Figures 6 and 7 are eye-catching, but not very informative. More detailed explanatory captions would help.
My overall impression of the explanations offered is that they are rather general and unconvincing. More detail is required.
In the Abstract, Z-scores and p-values are given without any basis in the article itself! The p-values are very high. Is this an error?
Comments on the Quality of English LanguageEnglish language deteriorates as the paper progresses.
Line 50: a word missing?
Line 63: what does ‘stamping out’ mean?
Line 69: ‘The stretching of a spastic and contracted muscle fibers and release substances’ - not a complete sentence.
Line 276: has been shown
Line 284. is as explained?
Line 331. singular/plural: 'there is no significant differences'
Author Response
Dear Reviewer,
We would like to express our sincere appreciation for the insightful comments and constructive suggestions provided regarding our manuscript entitled “Botulinum Toxin in Pain Related Post Stroke Limb Spasticity: A Meta-Analysis on Early and Late Injections”.
Your feedback has been extremely valuable in improving the clarity and overall quality of our work. We have carefully considered all points raised and have revised the manuscript accordingly. Here, we provide our responses to each reviewer, addressing their comments in a point-by-point manner.
We are grateful for the time and effort invested by the reviewers and yourself in the evaluation of our manuscript, and we hope that the revised version meets your expectations.
Thank you for your continued consideration.
Sincerely,
On behalf of all co-authors
Comments and Suggestions for Authors
Comment 1: [This is an interesting study, but has not been adequately proof read. For example, sometimes 'BoNT-A' is used, and sometime 'BoNT- A', and sometimes spacing between words is inconsistent (e.g., line 49)]
Response 1: Thank you for this reminder. This was addressed for the whole manuscript.
Comment 2: [Abbreviations and acronyms used are not always defined, and should be when first used. A list of these would help some readers.]
Response 2: This was addressed for the whole manuscript.
Comment 3: [Line 95: Were ONLY RCTs included? If so, please indicate.]
Response 3: This is indicated at Line 89-90
Comment 4: [For example, BoNT-A, HRQoL]
Response 4: This is addressed for the whole manuscript.
Comment 5: [Methodology is good (use of PEDro; careful exclusions); however, with only 3 studies for early BoNT-A, how valid are the results?]
Response 5: Yes, this is a declared limitation of the study. But these are the only available data set there is currenty.
Comment 6: [Lines 170-1: 'not statistically significant (Z = 3.90, 170 P <0.0001).' But p is very small, so how is this non-significant? It is significant with the p value of less than 0.05. Methods and software used for Chi2 and I2, as well as other statistical tests, should be more fully described. Was RevMan used, for example?]
Response: Yes, RevMan was used to determine the Chi2 and I2 as well as the forest plot. We also corrected the interpretation of p value at Line 177-179
Comment 7: [Figures 6 and 7 are eye-catching, but not very informative. More detailed explanatory captions would help.]
Response 7: We will provide more elaborate explanations as seen revised at Line 268-276 for Figure 6 and Figure 7 is further explained in Lines 295-300.
Comment 8: [My overall impression of the explanations offered is that they are rather general and unconvincing. More detail is required.]
Response 8: This is duly noted; further details added to the said figures.
Comment 9: [ In the Abstract, Z-scores and p-values are given without any basis in the article itself! The p-values are very high. Is this an error? It is specified at the tables provided ]
Response 9: We already corrected the p values/ z scores. The p values/ z scores are seen in the lower part of the Forrest plots.
Comments on the Quality of English Language
Comment 1: [ English language deteriorates as the paper progresses. ]
Response 1: Agree with this dear editor, Reediting of the entire manuscript was done.
Comment 2: [ Line 50: a word missing?]
Response 2: We already added the word tissue
Comment 3: [ Line 63: what does ‘stamping out’ mean?]
Response 3: We already replaced this with “contraction” instead of stamping as seen in Line 57.
Comment 4: [ Line 69: ‘The stretching of a spastic and contracted muscle fibers and release substances’ - not a complete sentence. ]
Response 4: We already corrected this at line 55 to 61; this is a redundant wording/phrasing for the previous statements.
Comment 5: [ Line 276: has been shown]
Response 5: This was corrected this at Line 281.
Comment 6: [ Line 284. is as explained?]
Response 6: This was corrected this at Line 289.
Comment 7: [ Line 331. singular/plural: 'there is no significant differences']
Response 7: This was corrected this at Line 325.

Round 2
Reviewer 2 Report
Comments and Suggestions for Authors
Thank you for your revisions. I am still unclear on one point: In the Abstract, you state "This study also concludes that BoNT- A injections are still preferred over placebo for the management of pain-related spasticity during the late phase [z = 1.48 (p = 0.14)]". However, in lines 213-14, you state "However, this effect was not statistically significant". Isn't this a contradiction?
Author Response
[Comment 1]: Thank you for your revisions. I am still unclear on one point: In the Abstract, you state "This study also concludes that BoNT- A injections are still preferred over placebo for the management of pain-related spasticity during the late phase [z = 1.48 (p = 0.14)]". However, in lines 213-14, you state "However, this effect was not statistically significant". Isn't this a contradiction?
Response 1: Thank you for your comment.
We will revise the statement in line 17-19
While BoNT-A appears somewhat more effective than placebo in the late phase, as indicated by the small to moderate positive z value, there is not enough evidence to confidently claim superiority over placebo [z = 1.48 (p=0.14)].
We will revise also in line 212-215
The overall mean difference between the experimental and control groups was 0.11 (95% CI: [-0.04, 0.26]) showing improvement in pPSS favoring BoNT-A injections with a small to moderate effect (z= 1.48). However, this effect was not enough to claim superiority (p = 0.14).
We will retain our statements in line 347-348
This study also concludes that during the late treatment period (>12 weeks), BoNT-A is still preferred over placebo for the management of pPSS.
